# Two-step synthesis of chiral fused tricyclic scaffolds from phenols via desymmetrization on nickel

Ravindra Kumar[1], Yoichi Hoshimoto [1,2], Eri Tamai[1], Masato Ohashi[1] & Sensuke Ogoshi [1]

Tricyclic furan derivatives with multiple chiral centers are ubiquitous in natural products. Construction of such tricyclic scaffolds in a stereocontrolled, step-economic, and atom-economic manner is a key challenge. Here we show a nickel-catalyzed highly enantioselective synthesis of hydronaphtho[1,8-*bc*]furans with five contiguous chiral centers via desymmetrization of alkynyl-cyclohexadienone by oxidative cyclization and following formal [4 + 2] cycloaddition processes. Alkynyl-cyclohexadienone was synthesized in one step from easily accessible phenols. This reaction represents excellent chemo-selectivity, regio-selectivity, diastereo-selectivity, and enantio-selectivity (single diastereomer, up to 99% ee). An extraordinary regioselectivity in the formal [4 + 2] cycloaddition step with enones revealed the diverse reactivity of the nickelacycle intermediate. Desymmetrization of alkynyl-cyclohexadienones via oxidative cyclization on nickel was supported by the isolation of a nickelacycle from a stoichiometric reaction. Enantioenriched tricyclic products contain various functional groups such as C=O and C=C. The synthetic utility of these products was demonstrated by derivatization of these functional groups.

[1] Department of Applied Chemistry, Faculty of Engineering, Osaka University, Suita, Osaka 565-0871, Japan. [2] Frontier Research Base for Global Young Researchers, Graduate School of Engineering, Osaka University, Suita, Osaka 565-0871, Japan. Correspondence and requests for materials should be addressed to S.O. (email: ogoshi@chem.eng.osaka-u.ac.jp)

Hydronaphtho[1,8-*bc*]furan rings with multiple chiral centers are a common structural motif in biologically active natural products (Fig. 1a)[1–5]. Such tricyclic structures are also found in key synthetic intermediates that are employed in a large number of sesquiterpenoids[6–9]. Owing to diverse biological activities and synthetic potentials associated with these fused tricyclic structures, a significant amount of attention has been paid to their enantioselective syntheses. Despite the existence of various stepwise stereoselective methods[1–9], direct access to such tricyclic fused rings in a completely enantio-controlled, diastereo-controlled, step-economic, and atom-economic manner would be a remarkable development[10–13]. Recently, Alemán reported a straightforward method for the construction of tricyclic fused rings from a cyclohexadienone tethered alkenal by employing an organocatalyzed asymmetric desymmetrization strategy (Fig. 1b)[14]. In this process, desymmetrization step involved the intramolecular [4 + 2] cycloaddition of chiral dienamine with a diastereotropic enone constructed tricyclic fused rings with three chiral centers.

We envisaged the enantioselective desymmetrization[15–19] of alkynyl-cyclohexadienone via an intramolecular oxidative cyclization on nickel in the presence of a chiral ligand. Alkynyl-cyclohexadienone were synthesized in one step from easily accessible phenols. The oxidative cyclization of an enantiotropic enone with a tethered alkyne unit would form a tricyclic fused nickelacycle[20–24] with three chiral centers, which could react with another olefin to yield a tricyclic product with the concomitant generation of two more chiral centers. Nickel(0)-catalyzed trimerization of an alkyne with two enones has been reported by Montgomery et al.[25–28] and by us[29, 30]. Here, we report a nickel(0)-catalyzed enantioselective synthesis of chiral hydronaphtho[1,8-*bc*]furans with five contiguous chiral centers via desymmetrization of alkynyl-cyclohexadienone and following intermolecular formal [4 + 2] cycloaddition reaction processes.

## Results

**Reaction optimization.** Prior to developing the reaction in an asymmetric fashion, achiral ligands were examined using cyclohexadienone-yne (**1a**) and 4-methoxychalcone (**2a**) for the model transformation to hydronaphtho[1,8-*bc*]furan **3aa** (Fig. 2, see also Supplementary Table 1 for detail). 1,3-Bis-(2,6-diisopropylphenyl)imidazol-2-ylidene (IPr) proved to be an optimal ligand to deliver *rac*-**3aa** in 96% yield, whereas PCy₃ failed to give any product. A single diastereomer of **3aa** was obtained out of sixteen possible isomers. Moreover, it is remarkable that an extraordinary regioselectivity was observed in the formal [4 + 2] cycloaddition step, fixing two carbonyls at the 1,4-positions in **3aa**, whereas 1,5-dicarbonyl compounds were obtained in reports of nickel(0)-catalyzed cycloaddition reactions[20–30]. Considering

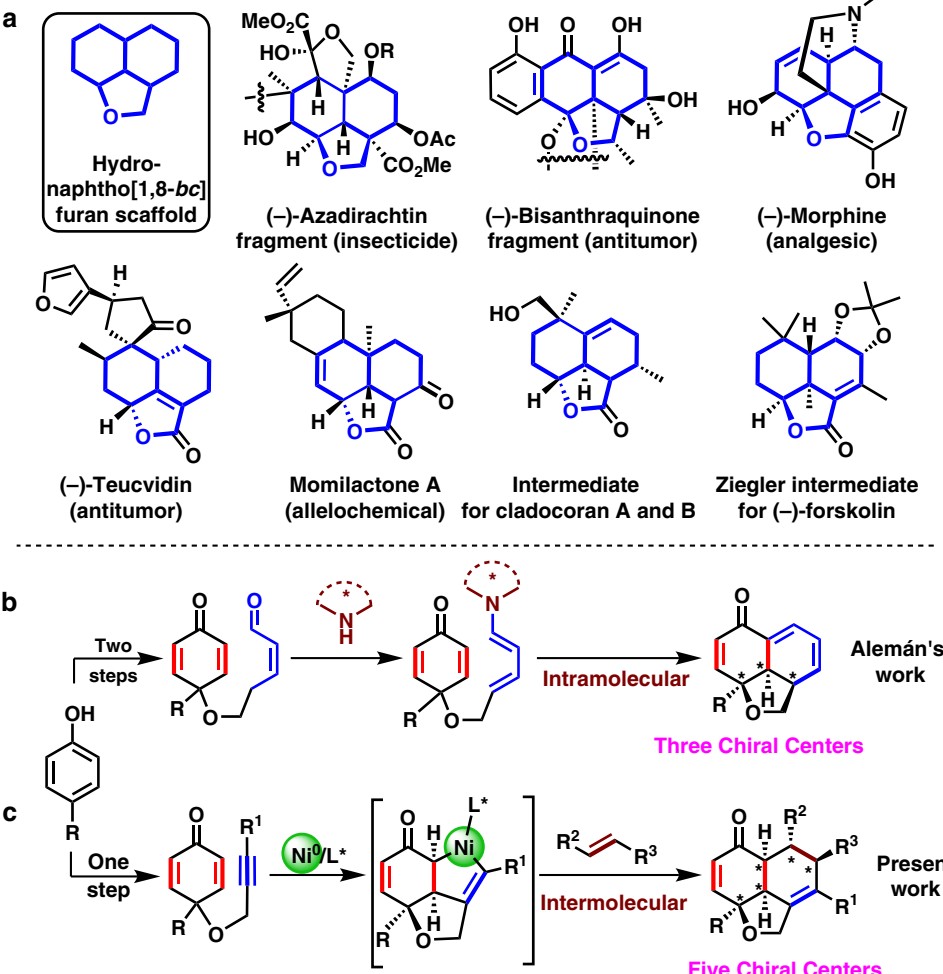

**Fig. 1** Hydronaphtho[1,8-*bc*]furans and their syntheses from phenols. **a** Representative examples of biologically active compounds. **b** Previous work employed desymmetrization by intramolecular [4+2] cycloaddition, catalyzed by organocatalyst. **c** Present work employed desymmetrization by oxidative cyclization and following intermolecular formal [4+2] cycloaddition process with nickel

**Fig. 2** Evaluation of Ligands. (*R,R*)-**L\*n**·HBF$_4$, NaO$^t$Bu, Ni(cod)$_2$ (0.01 mmol each), **1a** (0.12 mmol), **2a** (0.10 mmol), and toluene (1.0 ml) are employed. Isolated yields are given and enantioselectivity was determined by SFC equipped with a chiral stationary phase. [a]Yield and enantioselectivity was measured at 27% conversion in 3 days. [b]5 ml toluene was employed and reaction was conducted at 60 °C. [c]10 mol% IPr was used instead of chiral NHC salt and NaO$^t$Bu. [d]20 mol% PCy$_3$ was used instead of chiral NHC salt and NaO$^t$Bu

the efficiency of *N*-heterocyclic carbene (NHC) in this transformation, chiral (*R,R*)-NHCs, generated in situ by treating the corresponding imidazolinium salts with NaO$^t$Bu, were investigated to afford enantioenriched **3aa** (Fig. 2). It is worth mentioning that despite much exploration of the use of chiral NHCs, there has been less reports with nickel-catalyzed reactions[31–40]. *N*-(2-Biphenyl)- (**L\*1**) and *N*-(2-isopropylphenyl)- (**L\*2**) substituted NHCs[41] were ineffective to give **3aa**. In a similar manner, *N*-(2,7-diisopropylnaphthyl)- (**L\*4**)[42] and *N*-(2,7-dicyclohexylnaphthyl) (**L\*5**)[34]-substituted NHCs also failed to yield any products. However, *N*-2,6-diethylphenyl-substituted NHC (**L\*3**)[43] successfully gave hydronaphtho[1,8-*bc*]furan **3aa** in 13% yield with high enantioselectivity (94% ee). NHCs **L\*6**[35] and **L\*7**[34] furnished **3aa** in moderate chemical yields (36 and 48%, respectively) with excellent enantioselectivities (98 and 89% ee, respectively). Given the excellent enantioselectivity with **L\*6**, extensive effort was devoted to improving the yield. When the reaction was conducted at 60 °C using a lower concentration (0.02 M of **2a**), **3aa** was obtained in 74% yield with 98% enantioselectivity (see Supplementary Table 1 for detail).

**Substrate scope**. With the aforementioned optimal reaction conditions, we explored the scope of substrates (Fig. 3). A range of electron-rich and electron-deficient aryl-substituted enones **2** was examined with **1a**. The reaction proceeded smoothly with **2b** and **2c**, giving **3ab** and **3ac** in 73 and 72% yields, respectively, with enantioselectivities of 98% each. Reaction with an enone containing 2-furyl group (**2d**) was also examined with **1a** to afford **3ad** in 72% yield with 99% ee. In contrast, the reaction of **1a** with 1-aryl-2-buten-1-ones (**2e**–**2i**) gave **3ae**–**3ai** in slightly lower yields (62–70%), albeit enantioselectivities remained excellent (94–97% ee). In these cases, about 5% of fully-intermolecular [2 + 2 + 2] cycloaddition products (**3′**) of an alkyne unit of dienone-yne (**1a**) with two enones (**2e**–**2i**) were observed (See Supplementary Fig. 10 for **3ae′**). When ethyl group at R$^3$ of an enone was introduced, a complex mixture was obtained. Next, alkynyl-cyclohexadienone substrates (**1**) were investigated by varying the substituents R$^1$ and R$^2$. Ethyl as well as 2-methoxyethyl-substituted alkynyl-cyclohexadienone (**1b** and **1c**) gave **3bb**, **3bj**, **3ca**, and **3ck** with **2b**, **2j**, **2a**, and **2k**, respectively in good yields (60–73%) with excellent enantioselectivities (96–99%). Aryl and alkyl groups on alkynes were also examined. Phenyl-acetylene-substituted dienone **1d** gave **3de** in good yield with excellent enantioselectivity (70% yield and 99% ee). Electron-rich anisyl-group substrate **1e** gave **3ec** (71% yield and 98% ee) and **3ek** (68% yield and 99% ee) with *p*-halogenated (F– and Cl–) chalcones **2c** and **2k**, respectively. No corresponding dehalogenated products were detected. An alkynyl-dienone **1f** bearing an electron-deficient *p*-CO$_2$EtC$_6$H$_4$– group gave **3fc** with **2c** in 60% yield with 98% ee, whereas **3ge** was obtained in 71% yield with 96% ee from *p*-CF$_3$-substituted **1g**. An ethyl-group and a triethylsilyloxy-methyl-substituted alkynyl-cyclohexadienone (**1h** and **1i**, respectively) gave corresponding tricyclic fused rings **3he**, **3ic**, and **3ik** with **2e**, **2c**, and **2k**, respectively, in good yields and enantioselectivities (69–77% yields, 95–98% ee). A *N*-tosyl analog of alkynyl-cyclohexadienone **1j** failed to give the desired product **3jc** with **2c** under the present reaction conditions. This could have been due to coordination ability of tosyl group to nickel which inhibits the coordination of **2c**[44, 45]. The absolute configurations of all five chiral centers in **3** were assigned according to an analogy with **3ik**, which was unambiguously determined by X-ray crystallographic analysis (Fig. 3). It also supports all the stereo selective and regioselective outcomes in **3**.

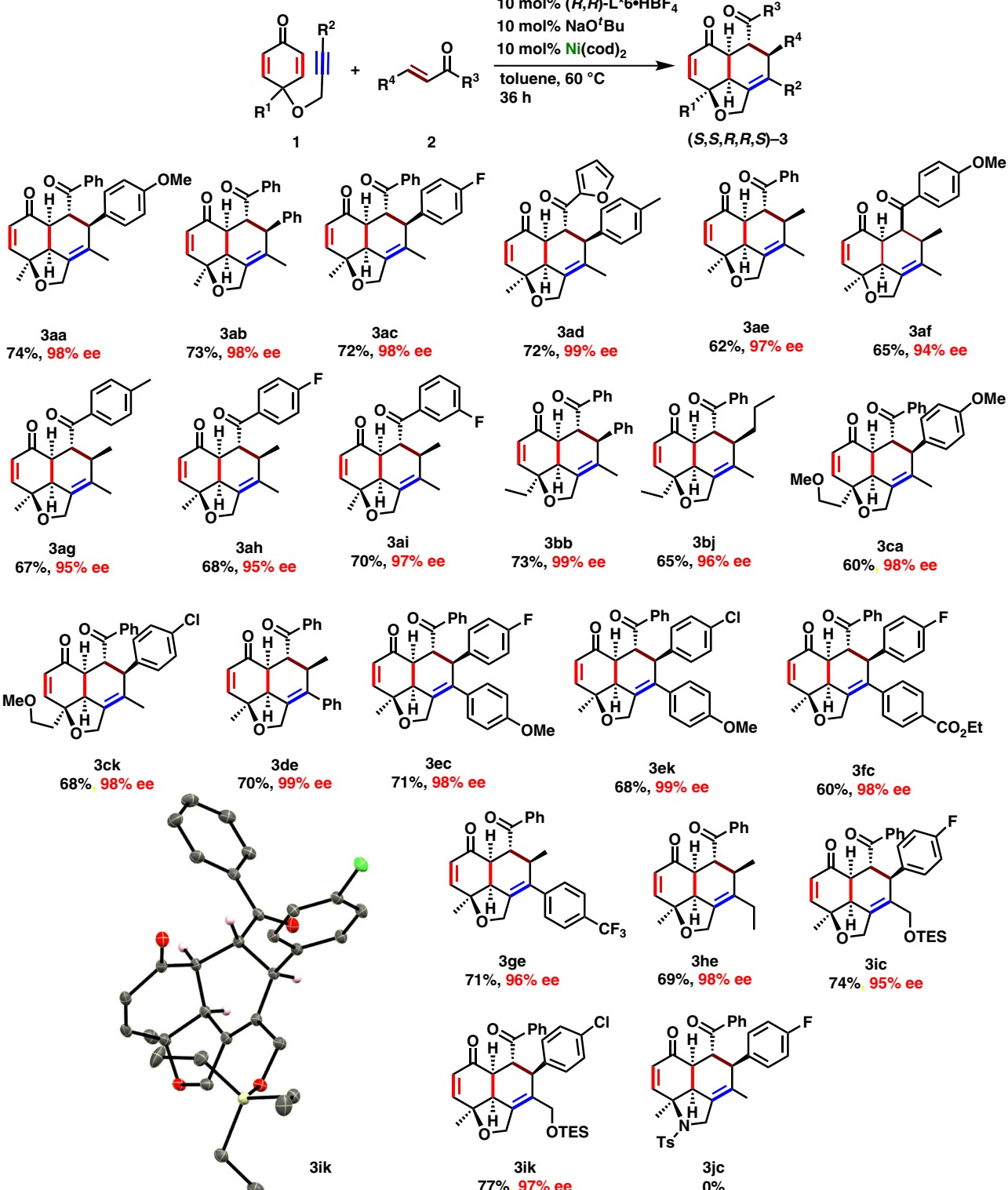

**Fig. 3** Substrate Scope. Reaction was examined at 0.1–0.4 mmol scale. Isolated yields are given and enantioselectivity was determined by SFC equipped with a chiral stationary phase. ORTEP diagram of **3ik** is shown with thermal ellipsoid at 30% probability level. Hydrogen atoms are omitted for clarity except those bound to the chiral carbon centers. Flack parameter for **3ik** = 0.017(7)

**Gram scale synthesis and transformations of product 3aa.** To demonstrate applicability, a half-gram-scale reaction of **1a** (0.53 g, 3.0 mmol) was carried out with **2a** (0.6 g, 2.5 mmol) to afford **3aa** in 73% yield and 98% ee. These enantioenriched tricyclic products could be useful synthetic intermediates for further transformations (Fig. 4). The methylene group of a tetrahydrofuran ring was oxidized with PCC[46] to yield a butyrolactone scaffold (**4**, 90% yield), that is present in natural products and also is a key synthetic intermediate in many sesquiterpenoids (Fig. 1a)[6–9]. Epoxidation and Michael addition of an enone gave the corresponding functionalized products **5** and **6** as single diastereomers in 75 and 85% yields, respectively, whereas hydrogenation of an

**Fig. 4** Synthetic Transformation of **3aa**. Isolated yields are given and enantioselectivity was determined by SFC equipped with a chiral stationary phase

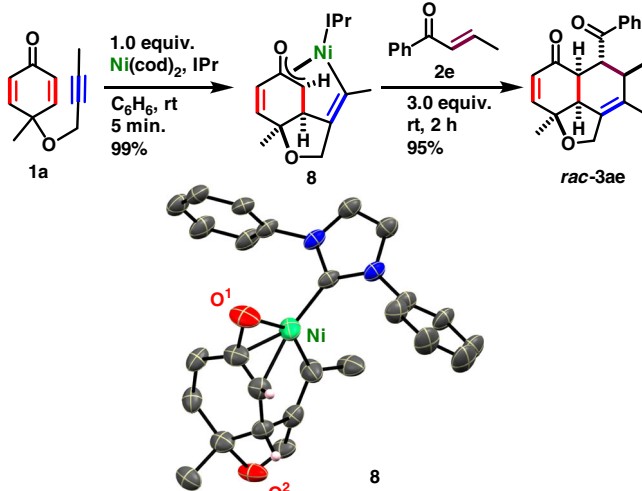

**Fig. 5** Stoichiometric reaction with **1a**. Isolated yield of **8** and *rac*-**3ae** are given. The molecular structure of η³-cyclohexadienyl Ni-complex **8** is disordered and one of the disordered structure is depicted for simplicity with thermal ellipsoid at 30% probability level. ${}^{i}$Pr groups and hydrogen atoms (except those bound to the chiral carbons) are omitted for clarity

enonic C = C bond with $H_2$/Pd(C) in ethyl acetate gave **7** in 89% yield. The enantioselectivity remained consistent in all these transformations.

**Stoichiometric experiment**. In order to gain deeper insight into a possible reaction mechanism, stoichiometric experiments were conducted in $C_6D_6$. An attempts to isolate a chiral nickelacycle corresponds to **1g** using an optimal NHC **L\*6** and Ni(cod)₂ was unsuccessful. [19]F NMR spectra showed seven peaks, revealed the existence of several intermediates, in which one of them might be much more reactive leads to desired product in the presence of an enone. However, an η³-oxaallyl nickelacycle (**8**) was isolated in 99% yield when a stoichiometric reaction of **1a** was conducted with IPr and Ni(cod)₂. The molecular structure of **8** was confirmed by X-ray crystallography (Fig. 5). The [1]H, [13]C, and 2D NMR analyses of **8** demonstrated that its structure in solution

was consistent with that observed in crystal lattice. The reaction of **8** with **2e** gave *rac*-**3ae** in 95% isolated yield, which supported that desymmetrization by oxidative cyclization would play a key role in the present transformation.

A plausible reaction mechanism is drawn on the basis of the results of the stoichiometric experiment and previous reports (Fig. 6)[20–30, 33–42]. First, the intramolecular oxidative cyclization of **1** via the simultaneous coordination of an alkyne and an enantiotropic enone to the chiral Ni(0)/L\* species gives a desymmetrized nickelacycle **A**, which would be in equilibrium with its η³-oxaallylnickel structure **A′**. Coordination of **2** to nickel center of **A** giving **B**, followed by insertion through a Ni–$C_{sp^2}$ bond could form a thermodynamically favorable η³-oxaallylnickel structure either **C** or **C′**. Then, a subsequent reductive elimination could afford a tricyclic fused structure **3** as a single diastereomer with the regeneration of nickel(0) species.

In conclusion, a catalytic enantioselective method has been developed for the rapid construction of hydronaphtho[1,8-*bc*] furans with five contiguous chiral centers via desymmetrization of alkynyl-cyclohexadienone and formal [4 + 2] cycloaddition reaction with nickel. The synthetic utility of tricyclic products was also demonstrated. Isolation of desymmetrized η³-oxaallyl nickelacycle and subsequent reactions in the stoichiometric experiment revealed that desymmetrization by oxidative cyclization is the key in this transformation. Furthermore, unusual regioselectivity in the insertion step of enone revealed the diverse reactivity of an η³-oxaallyl nickel-complex. The developed strategy involving two steps from the easily accessible phenols, demonstrates a practical and step economic protocol to access synthetically valuable fused tricyclic frameworks bearing five consecutive chiral carbon centers with excellent enantioselectivities.

## Methods
**General procedure for tricyclic product 3**. To a screw cap vial in a glove box was added **L\*6**·HBF₄ (10 mol%) and NaO${}^t$Bu (10 mol%) and toluene (5 ml). The suspension was allowed to stir at room temperature for 10 min and then Ni(cod)₂ (10 mol%) was added. After further stirring for 10 min at room temperature was added a solution of alkynyl-cyclohexadienone (**1**, 0.24 mmol) and enone (**2**, 0.20 mmol) in toluene (5 ml). The reaction mixture was taken out of glove box and heated at 60 ° C for 36 h with stirring. After cooling to room temperature, the mixture was filtered through celite and washed with Et₂O. The filtrate was concentrated in vacuo and the residue was purified by silica gel flash chromatography (5–20% ethyl acetate in hexane) to afford the desired product **3**.

**Fig. 6** A plausible reaction mechanism. Nickelacycle **A** is generated by enantioselective desymmetrization of **1** via oxidative cyclization. Subsequent insertion of an enone **2** to **A** gives a tricyclic product **3**

**Data availability**. The X-ray crystallographic coordinates for structures reported in this study have been deposited at the Cambridge Crystallographic Data Centre (CCDC) under deposition numbers CCDC 1523827 (**3ik**) and 1523828 (**8**). These data can be obtained free of charge from The CCDC via www.ccdc.cam.ac.uk/data_request/cif. All other data supporting the findings of this study are available within the article and its Supplementary Information file or from the authors upon reasonable request. For NMR spectra of the compounds in this article, see Supplementary Figs. 2–31.

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

## Acknowledgements

This work was supported by a Grant-in-Aid for Scientific Research (A) (No. JP25708018) and (B) (No. JP15K17824), Scientific Research on Innovative Areas (Nos. JP15H00943 and JP15H05803) from MEXT and by JST, Advanced Catalytic Transformation Program for Carbon Utilization (ACT-C, Grant Number JPMJCR12Y6), Japan. Y.H. acknowledges support from the Frontier Research Base for Global Young Researchers, Osaka University, on the program of MEXT. We sincerely thank Prof. Dr. Norimitsu Tohnai, Graduate School of Engineering, Osaka University, Japan for collecting X-ray data of **3ik**. We also thank Prof. Dr. Mary Grellier, Laboratoire de Chimie de Coordination, CNRS, Toulouse, France for discussion on mechanistic part.

## Author contributions

S.O. and R.K. conceived and designed the synthetic routes. R.K. and Y.H. prepared the manuscript, and edited by all other authors. R.K. and E.T. carried out the experiments. M.O. analyzed the X-ray data.

## Additional information

**Competing interests:** The authors declare no competing financial interests.

