## [Peer Review File · Nature Communications]

Reviewers' comments:

Reviewer #1 (Remarks to the Author):

The hydronaphtho[1,8-bc]furan scaffold is a common structural motif of important natural products with powerful biological and pharmaceutical properties. In this manuscript, Ogoshi discloses an elegant method using nickel(0) catalysts to access this scaffold from very simple starting materials, in a fully controlled manner. Particularly, the control of enantioselectivity represents a tough challenge for modern nickel catalysis. Many nickel-catalyzed transformations work best (in terms of yield) with the achiral IPr carbene ligand outcompeting other carbenes as well as phosphine ligands.

Unfortunately, a simple one-fits all plug-and-play chiral IPr does not exist! Chiral carbene ligands are generally well established and many have been reported, with few providing high enantioselectivities. However, this is not true for Ni(0)-catalyzed reactions. Here, most do not provide reactivity and/or selectivity when used to render nickel complexes chiral. Developing a suitable chiral NHC for Ni is still significant and tough challenge. Despite the high relevance that these processes would have, only very few groups (Ogoshi, Montgomery, Cramer...) managed to successfully find solutions. The presented results constitute a very good advance and provide an elegant route to a complex scaffold with full control over 5 stereogenic centers. Not just a single stereocenter, but five in total! The scope of the transformation is adequately demonstrated concerning functional group tolerance and variability. Throughout high yields and excellent enantioselectivities were obtained. A point of criticism would be the variability of R3. Are alkyl groups not tolerated here? A furyl group could be a great surrogate that could be later degraded fairly easy into a carboxylic acid. Such substrate(s) should be added or the limitation briefly discussed.

The manuscript has a clear structure and is easy to follow. Few easy points to address:

- Some sentences (especially in the abstract „... various functionalities (enone, carbonyls, and C=C) ...”) should be rephrased for a better accessibility to non-organic chemists.
- The authors use “tandem desymmetrization”. The word “tandem” should be omitted as there is only a single desymmetrization event: which double bond of the dienone reacts.
- The resolution of Ortep of 3ij and 8 is low and should be improved.
- Figure 4: Here, simple methyl substituents are used on the structures to depict the mechanism. While probably used to simplify, it creates confusion. For instance the molecule tagged as 1 (of fig 4) is 1a in fig 3 and compound 2 implies that it would be an actual substrate. I recommend to use generic R's like for the scheme of table 2. Same for GA.
- the supporting information is generally of very high quality and carefully assembled. However, few 13C spectra as well as SFC traces are very noisy. They are still acceptable, but I would recommend replacing them with better ones. The SFCs for 4-7 could be removed (together with the notion in the caption of figure 2). There is no possibility for a racemization of 3aa. However, given a dr for 5 and 6 would be useful.

Overall, this is a fine manuscript, and I fully support publication upon addressing the abovementioned minor points.

Nicolai Cramer

Reviewer #2 (Remarks to the Author):

Ogoshi et al describe in this work the synthesis of tricyclic derivatives under Nickel asymmetric catalysis. In this process, the desymmetrization of alkynyl-cyclohexadienones by oxidative cyclization is taking place, and products with five chiral centers are formed with excellent enantioselectivities. A mechanism proposal based on the isolation of intermediate 8 is shown in Figure 4.

The authors prepared a large number of products (table 2), and some derivatizations in Figure 2. A large number of natural products are presented (top figure 1). However, the authors did not

synthesize them using this new methodology. In addition, there is a lack of novelty of this work, because related papers (e.g. ACIE 2014,8184) have been reported for the synthesis of these tricyclic derivatives in good enantioselectivities. For these reasons, I consider that this paper will not impact for a journal such as nature communications and it should be published in a more specialized journal.

Reviewer #3 (Remarks to the Author):

This manuscript from Ogoshi demonstrates a nickel-catalyzed 2+2+2 cycloaddition that allows desymmetrization of hexadienone substrates. The reaction has several novel features, as it is the first example of a desymmetrization process using this reaction type. The ee's are quite impressive, and the levels of enantioselectivities surpass that seem in most types of nickel-catalyzed pi-component couplings. Significant structural insights are provided into the operative reaction intermediates, and a logical mechanism is proposed. A reasonable scope of hexadienones are tolerated, although the authors should report at least one example with an enone that lacks a phenyl substituent at the ketone. Even if this outcome is not successful, it would be informative to the reader. The SI contains complete and careful characterization of products.

Overall, I think this manuscript is well executed and has interesting and novel findings. As such I consider it to be satisfactory for appearance in this journal.

Addition and Correction for the Manuscript

Response to referee 1

[1] A point of criticism would be the variability of R³. Are alkyl groups not tolerated here?

Ans: Thank you for bringing this point to our attention. We attempted the catalytic reaction of **1a** with (*E*)-hex-4-en-3-one (R³ = ethyl) under the optimized reaction conditions using 10 mol% Ni(cod)₂/L*6/NaO^tBu. It gave a complex mixture including a trace amount of the target product, which was confirmed by GC mass analysis. The statement is mentioned in page 5

[2] A furyl group could be a great surrogate that could be later degraded fairly easy into a carboxylic acid.

Ans: Thank you for suggesting a furyl containing compound. We examined enone **2d** containing 2-furyl group at the R³ position, which gave the target trimerization product **3ad** in 72% yield with 99% ee (Table 2, statement in page 7). The NMRs and SFC charts have been shown in Supplementary Information (page S10, S43–45)

[3] Some sentences (especially in the abstract „... various functionalities (enone, carbonyls, and C=C) ...”) should be rephrased for a better accessibility to non-organic chemists.

Ans: The sentence has been modified as follows:

"Enantioenriched tricyclic products contain various functional groups such as C=O and C=C. The synthetic utility of these products was demonstrated by derivatization of these functional groups."

[4] The authors use “tandem desymmetrization”. The word “tandem” should be omitted as there is only a single desymmetrization event: which double bond of the dienone reacts.

Ans: We would like to describe the sequential desymmetrization by oxidation cyclization and formal [4+2] cycloaddition steps as a tandem process. However, for clarity, I agree with the suggestion of the referee and the words "tandem" was omitted from the manuscript and the sentences were modified as indicated in manuscript (Abstract, page 1 and footnote of Fig. 2, page 3).

[5] The resolution of Ortep of **3ik** and **8** is low and should be improved.

Ans: Our attempts to improve the resolution of ORTEP diagrams of **3ik** and **8** using the Mercury software were unsuccessful due to its resolution limitation.

[6] Figure 4: Here, simple methyl substituents are used on the structures to depict the mechanism... I recommend to use generic R's like for the scheme of table 2.

Ans: The Fig. 4 has been modified by adding generic R's on the substrates to avoid confusion.

[7] The supporting information is generally of very high quality and carefully assembled. However, few ¹³C spectra as well as SFC traces are very noisy. They are still acceptable, but I would recommend replacing them with better ones.

Ans: All these compounds are pure and thus the apparent noise couldn't be removed after multiple attempts at purification. However, according to the suggestions, the SFC

charts of **3bj**, **3ck**, **3ec**, and **3ek** have been replaced with better ones (Supplementary Information).

[8] The SFCs for 4-7 could be removed (together with the notion in the caption of figure 2)However, given a dr for 5 and 6 would be useful.

Ans: We do agree with the referee's remark. There is no possibility for the racemization of **3aa** during transformation. Regardless, we measured it to ensure that. According to the referee's suggestion, the SFC charts for 4–7 have been removed. Single diastereomers were obtained in the cases of **5** and **6**, thus a statement was incorporated in the manuscript (page 7).

Response to referee 2

[1] The authors prepared a large number of products (Table 2), and some derivatizations in Figure 2. A large number of natural products are presented (top figure 1). However, the authors did not synthesize them using this new methodology.

Ans: Thank you for urging us to provide a complete demonstration of developed methodology for the total synthesis of chiral complex molecules. However, in this manuscript, we would like to reveal the development of the methodology for the efficient construction of fused tricyclic frameworks with five chiral centres.

[2] In addition, there is a lack of novelty of this work, because related papers (e.g. ACIE 2014,8184) have been reported for the synthesis of these tricyclic derivatives in good enantioselectivities.

Ans: Indeed, the above-mentioned reference describes an elegant method for the construction of tricyclic derivatives, our developed methodology is entirely different on the following points:

- a) The previous method involved a proline-based organocatalyst, whereas the present development is based on transition metal catalysis.
- b) The previous method generated tricyclic compounds with three chiral centers, whereas the present methodology produces tricyclic compounds with five chiral centres.
- c) In the previous method, the desymmetrization steps involved the asymmetric [4+2]

cycloaddition of diastereotropic enone and dienamine. In contrast, in the present work, the desymmetrization step involved oxidative cyclization on nickel. This is the first report of desymmetrization by oxidative cyclization on a metal.

Response to referee 3

[1] A reasonable scope of hexadienones are tolerated, although the authors should report at least one example with an enone that lacks a phenyl substituent at the ketone. Even if this outcome is not successful, it would be informative to the reader.

Ans: Thank you for bringing this point to mention in the manuscript. We attempted the catalytic reaction of **1a** with (*E*)-hex-4-en-3-one ($R^3 = \text{ethyl}$) under the optimized reaction conditions using 10 mol% $\text{Ni}(\text{cod})_2/\mathbf{L}^*6/\text{NaO}^t\text{Bu}$. It gave a complex mixture including a trace amount of target product, which was confirmed by GC mass analysis. The statement is mentioned in page 5.

REVIEWERS' COMMENTS:

Reviewer #1 (Remarks to the Author):

The authors have addressed my all of previously raised points. Therefore, I support publication of this manuscript.

- L22: change "derivertization" to "derivatization"

- Concerning the resolution of the X-ray images: We sometimes use POV-ray saving option from Mercury to create a .pov output file and then render the image with POV-ray v3.xx (allows for res up to 3200X2400).

http://bip.weizmann.ac.il/course/structbioinfo/databases/CCDC_Mercury/mercury.3.194.html#291258

Addition and Correction for the Manuscript

For Reviewer 1:

1) L22: Change “derivertization” to “derivatization”

Ans: We corrected “derivertization” to “derivatization” in line 22, page 1.

2) Concerning the resolution of the X-ray images: We sometimes use POV-ray saving option from Mercury to create a .pov output file and then render the image with POV-ray v3.xx (allows for res up to 3200X2400). http://bip.weizmann.ac.il/course/structbioinfo/databases/CCDC_Mercury/mercury.3.194.html#291258.

Ans: Thank you for your suggestion to use POV-ray technique to increase the resolution on X-ray images. We used it, and worked well. Higher resolution images of ORTEP diagrams of **3ik** and **8** are incorporated at appropriate places (Table 2 and Figure 3, respectively).